# Keep Calm and Carry on with Extra Centrosomes

**DOI:** 10.3390/cancers14020442

**Published:** 2022-01-17

**Authors:** Batuhan Mert Kalkan, Selahattin Can Ozcan, Nicholas J. Quintyne, Samantha L. Reed, Ceyda Acilan

**Affiliations:** 1Graduate School of Health Sciences, Koç University, Istanbul 34450, Turkey; bkalkan@ku.edu.tr; 2Research Center for Translational Medicine (KUTTAM), Koç University, Istanbul 34450, Turkey; cozcan@ku.edu.tr; 3Department of Biology, The State University of New York at Fredonia, Fredonia, NY 14063, USA; Nicholas.Quintyne@fredonia.edu (N.J.Q.); reed7151@fredonia.edu (S.L.R.); 4School of Medicine, Koç University, Istanbul 34450, Turkey

**Keywords:** cancer, centrosome, clustering, multipolar spindles, chromosomal instability

## Abstract

**Simple Summary:**

Precise chromosome segregation during mitosis is a vital event orchestrated by formation of bipolar spindle poles. Supernumerary centrosomes, caused by centrosome amplification, deteriorates mitotic processes, resulting in segregation defects leading to chromosomal instability (CIN). Centrosome amplification is frequently observed in various types of cancer and considered as a significant contributor to destabilization of chromosomes. This review provides a comprehensive overview of causes and consequences of centrosome amplification thoroughly describing molecular mechanisms.

**Abstract:**

Aberrations in the centrosome number and structure can readily be detected at all stages of tumor progression and are considered hallmarks of cancer. Centrosome anomalies are closely linked to chromosome instability and, therefore, are proposed to be one of the driving events of tumor formation and progression. This concept, first posited by Boveri over 100 years ago, has been an area of interest to cancer researchers. We have now begun to understand the processes by which these numerical and structural anomalies may lead to cancer, and vice-versa: how key events that occur during carcinogenesis could lead to amplification of centrosomes. Despite the proliferative advantages that having extra centrosomes may confer, their presence can also lead to loss of essential genetic material as a result of segregational errors and cancer cells must deal with these deadly consequences. Here, we review recent advances in the current literature describing the mechanisms by which cancer cells amplify their centrosomes and the methods they employ to tolerate the presence of these anomalies, focusing particularly on centrosomal clustering.

## 1. Introduction

Centrosomes are the major microtubule-organizing centers of the animal cell and are primarily known for their role in the regulation of microtubule nucleation [1]. Centrosomes also regulate other microtubule-dependent cell processes, such as cell shape and polarity in interphase cells [2]. Additionally, centrosomes function as a basal body for the primary cilium in quiescent cells [3].

Animal cells normally contain one or two centrosomes each, depending on their progression through the cell cycle. Mitosis generates two daughter cells, each possessing one centrosome inherited from the mother cell. A mature and functional centrosome is composed of two centrioles, a mother and a daughter, surrounded by pericentriolar material (PCM). During the S phase, centriole duplication is initiated on the wall of the mother centriole, while additional PCM components are trafficked to the centrosome via microtubules in a cytoplasmic dynein-dependent process [4], though some components can be recruited in a microtubule-independent manner [5]. This generates two centrosomes, which have a pair of centrioles surrounded by PCM. At the end of G2, phosphorylation of linker proteins that bridge between mother centrioles allows the centrosomes to be separated. Subsequently, the two centrosomes serve as the core of the mitotic spindle poles, guiding bipolar and symmetrical division to ensure even distribution of genetic material [6]. Disruptions in any one of these events may lead to centrosomal anomalies [7] that are associated with various physiological conditions or diseases, including reproductive disorders, ciliopathies, or cancer, which is the focus of this review.

Due to a variety of factors, tumor cells frequently exhibit extra centrosomes (greater than two), a state often referred to as centrosome amplification (CA), which has been reported in a wide array of human cancers including solid tumors of various origins and hematological malignancies [8]. CA has long been considered a contributing factor to chromosomal instability (CIN) [9] and even proposed as a biomarker for personalized therapy for some types of cancers [10]. Frequently, extra centrosomes lead to the formation of multipolar spindles (MPS) during mitosis. As a consequence of a multipolar division, the daughter cells are likely not viable, since significant amounts of genetic material will fail to segregate properly and the cell will not inherit a full copy of the genome. Yet, cancer cells are able to circumvent MPS formation by clustering extra centrosomes in two poles, enabling bipolar division. Accumulated research has improved our understanding that the centrosome-clustering mechanism is a key defense against cell death induced by multipolar division. Through clustering extra centrosomes, cancer cells can survive the potential consequences associated with CIN. In this review, we outline the control of centrosome biology in normal cells, continue with changes in tumor biology that affect centrosome number, and conclude by discussing contributions of CA to cancer cell biology.

## 2. Control of Centrosome Biology

The centrosome duplication cycle is a heavily regulated intracellular process occurring during the cell cycle at roughly the same time as DNA replication. The regulation of the centrosome duplication cycle ensures that the centrosome is duplicated only once per cell cycle, in parallel to limiting DNA replication to a single occurrence per cell cycle. Coordination of these events is linked with the activity of cyclins and cyclin-dependent kinases (CDK) [11]. Protein levels of cyclins fluctuate throughout the cell cycle; for example, cyclin E levels peak at the G1-S transition [12], whereas expression of cyclin A and cyclin B is maximal during G2 and M phases [13]. When the intracellular levels of cyclins are sufficiently high, they activate their corresponding CDKs and form cyclin/CDK complexes. Given that the initiation of centrosome duplication occurs at late G1-S, it is not surprising to find out that the cyclin E/CDK2 complex regulates this event [11]. Both centrosome duplication and DNA replication are under the control of CDK2, which is activated by hyperphosphorylation of the retinoblastoma protein (pRB) [14]. As centrosome duplication is initiated, cyclin E/CDK2 phosphorylates nucleophosmin, inducing its dissociation from the centrosome. This process prevents premature centriole splitting, and a nonphosphorylatable form of nucleophosmin was demonstrated to be sufficient to prevent centrosome duplication [15]. As centrosome duplication progresses, in addition to the cyclin E/CDK2 complex, there is a requirement for activity and regulation by cyclin A/CDK2 [16]. Subsequently as centrosome duplication enters its later stages, dissolution of centriole linkers at G2/M transition is controlled by cyclin B/CDK1-mediated phosphorylation of Eg5 and Nek2-mediated phosphorylation of centriole linker proteins [17,18,19,20]. Further activity of cyclin/CDK complexes in this process can be seen because centrosome duplication in somatic animal cells requires the phosphorylation of pRB. Overexpression of E2F, a downstream effector of pRB, was found to be sufficient to override the effect caused by the expression of a non-phosphorylatable mutant of pRB, indicating E2F is the major effector of pRB pathway on centrosome duplication [16].

Duplication of centrosomes is also controlled intra-centrosomally. The Polo-like kinase PLK4 is known as the master regulator of centrosome duplication [21]. PLK4 is required for the biogenesis of a new pro-centriole [22]. In addition, this study also identified several other proteins necessary for procentriole formation: HsSAS-6, CPAP, CEP135, CP110, and γ-tubulin are required at different stages of procentriole formation [22]. Other studies have identified more key players: STIL contributes to centriole biogenesis via interplay with PLK4. PLK4 binds to and activates STIL by phosphorylation [23], and vice versa [24,25,26]. Importantly, the coiled-coil domain of STIL stabilizes PLK4, whereas the C-terminal domain of STIL removes excessive PLK4 activity, ensuring the formation of a single procentriole perpendicular to the wall of the mother centriole [27]. PLK4 activity is also regulated by auto-phosphorylation [28,29,30]. Independent studies have demonstrated that the interplay between PLK4 and STIL is the key to proper centriole biogenesis; this interaction is crucial for centriolar recruitment of HsSAS6 [26]. Notably, depletions of PLK4, STIL, or HsSAS-6 have been shown to block centrosome duplication [31,32,33]. Furthermore, CEP63 plays an early role in centriole biogenesis, preceding recruitment of HsSAS-6 [34]. Other CEP family proteins, CEP152 and CEP192, also interact with PLK4, and the conversion of PLK4 from a CEP192-tethered-and-sequestered state to a CEP152-bound-procentriole-assembly state is a critical requirement for centriole duplication [35,36].

Another feature of centrosome biology is centriole elongation (CE), which is initiated after procentriole formation. CE is regulated by several factors including CPAP, CP110, and CEP120, and is controlled by their activities and interactions. Centrosomal protein 4.1-associated protein (CPAP) is required to incorporate microtubules after pro-centriole assembly [37], and CP110 has been identified as a distal end-capping protein [38,39]. In addition, CEP120 interacts with CPAP and positively regulates CE [40]. A study using proteasome inhibitors demonstrated that inhibition of cellular proteolysis increases centriole length: observed centrioles were approximately four times normal length, showing that proper CE is dependent upon normal cellular proteolysis [41]. In this study, a siRNA screen uncovered several key centrosomal proteins. In addition to the aforementioned CPAP and CP110, seven other proteins were identified as CE regulators: FOP, CAP350, HsSAS-6, CEP170, ninein, C-Nap1, and CEP97 [41].

Since centrosome biology is regulated by many factors, dysregulation of any of them could cause centrosomal structural defects or aberrant centrosome numbers, leading to aneuploidy and chromosomal instability. Given that CA is a common feature of tumors, it is important to reveal the underlying mechanisms that cause CA.

## 3. Centrosome Amplification in Tumors and the Causes

Centrosome amplification in human cancers was first described by Boveri [42] more than a century ago and has gained attention over eighty years later when the discovery of the relation between the loss of p53 and centrosome amplification was made [43,44]. Subsequent studies added to this finding and researchers began to understand the causes of centrosome amplification in cancers better. Supernumerary centrosomes were observed in many clinical tumor samples [8], precancerous lesions [45], and cancer cell lines [46]; CA is also associated with poor patient prognosis [8]. Identification of the specific mechanisms of CA in cancers may help suppress cancer growth with inhibitors targeting CA mechanisms.

The question has always been whether CA was a cause or consequence of cancer, and some recent research has shown that under some specific conditions, amplification of centrosomes can promote tumorigenesis and tumor invasion [47]. One study has demonstrated that CA can initiate tumorigenesis in flies [48]. Similarly, a more recent study found that elevation of centrosome numbers increased tumor initiation in a mouse intestinal tumor model [49]. This study also showed the presence of supernumerary centrosomes was sufficient to drive aneuploidy and the development of spontaneous tumors in multiple tissues [49]. CA is also associated with increased invasiveness of tumors, and a study strikingly found that cells with extra centrosomes could induce invasion of neighboring cells in breast cancer via paracrine signaling, and secreted IL-8 was identified as a crucial factor for the induction of invasion via HER2 activation [50]. This study suggests that while only a small proportion of cells within a heterogeneous tumor mass may possess supernumerary centrosomes, these cells can exert a disproportionate effect upon tumor progression. Given that CA is increasingly associated with multiple aspects of tumor biology, it is important to understand the mechanisms that cause CA in tumors.

Mechanisms of centrosome amplification in cancers can be divided into two major classes: (i) overproduction of centrosomes due to the loss of tight control over the centrosome duplication process arising from pathway-inactivating genetic mutations and/or overexpression of centrosome duplication factors (direct centrosome amplification); and (ii) other cell biological events, which cause relative centrosome amplification, such as cell-cell fusions, failures of cytokinesis, premature centriole disengagement or centrosome (PCM) fragmentation (indirect centrosome amplification) (Summarized in Figure 1).

### 3.1. Direct Centrosome Amplification

The major route for the presence of supernumerary centrosomes in cancers is the deregulation of the centrosome duplication cycle. Many positive and negative regulators contribute to the control of the centrosome cycle, including tumor suppressors, such as p53, pRB, PTEN, or BRCA1. Loss of function mutations of these genes are associated with CA in various types of tumors. In addition, misexpression of proto-oncogenes, including E6, E7, Myc, and Ras is also associated with CA. Furthermore, changes in the genetic and epigenetic background in cancers can lead to the overexpression of centrosome duplication factors, such as PLK4, STIL, and HsSAS-6, which leads directly to CA by overproducing new centrioles.

#### 3.1.1. Tumor Suppressors and Proto-Oncogenes

p53 is a well-defined tumor suppressor transcription factor that causes cell cycle arrest or apoptosis in response to DNA damage or other cellular stresses. In addition to being a tumor suppressor transcription factor, p53 also localizes to centrosomes, and the timing of the association of p53 with the centrosome is suggested to be an important regulatory step during mitosis [51,52]. Two main regulatory processes of centrosome duplication involve control by p53: initiation of centrosome duplication during late G1 [53], and suppression of a second duplication event during S and G2 [44]. Loss of p53 is one of the well-characterized causes of centrosome amplification in cancers [43]. Under p53^−/−^ conditions, overexpression of p21^Waf1/Cip1^ was shown to partially revert the CA phenotype and reintroduction of wild type p53 completely reverted the p53^−/−^ effects; suggesting p53 regulates centrosome duplication through multiple pathways, including a pathway where p21 functions as the effector protein [44,54]. Moreover, constitutive activation of the Cyclin E/CDK2 complex resulted in the uncoupling of centrosome duplication from DNA replication in MEFs derived from a p53 null mouse, and as a potent inhibitor of the Cyclin E/CDK2 complex [55], p21 was found to be pivotal in the coordination of these two key cell biological events [56]. Furthermore, both CDK2 and CDK4 have been identified as critical mediators of CA caused by loss of p53 [57]. Other proteins that are key players in cell cycle progression and inhibition are also critical to regulation of centrosome duplication; overexpression of the cell cycle inhibitory proteins, p16^INK4A^, p21^Waf1^ and p27^Kip1^, will block centrosome duplication [14,16,58]. Additionally, GADD45, a target of p53, has also been identified as a contributor to the regulation of centrosome duplication: GADD45^−/−^ mice exhibit many of the characteristics observed in p53^−/−^ mice including CA [58].

Whereas the loss or inactivation of p53 alone is sufficient to induce centrosome amplification in mouse cells, p53 silencing in human cells is not enough for the generation of CA, indicating additional factors are necessary. Numerous studies have given us clues about some of these factors that can work in conjunction with a downregulation of p53 functionality to drive CA in humans. One study identified cyclin E overexpression as a requirement to achieve CA by p53 inactivation in human bladder cancer cells [59]. Another demonstrated that overexpression of Aurora A kinase induces CA and aneuploidy in not only mouse NIH 3T3 cells, but also human breast cancer cells [60]. Given that Aurora A phosphorylates p53 at S315 and marks it for MDM2-dependent degradation [61], it is thought that Aurora A at least partially contributes to CA by driving p53 inactivation [62]. However, Aurora A overexpression, along with that of other mitotic kinases, can also contribute to both CA and polyploidy via another method: in a p53^−/−^ background, overexpression of these kinases indirectly led to CA not through errors or deregulation of the centrosome duplication machinery, but instead by permitting cell viability to persist after defects in cytokinesis [63,64,65]. A more recent study also demonstrated that altered expression of NDRG1 (N-Myc down-regulated gene 1), a p53-inducible gene, affected centrosome number. NDRG1 was shown to directly interact with γ-tubulin and this interaction was reduced significantly in p53-knockout cells. This interaction and the function of NDRG1 was characterized as a pivotal component for normal centrosome homeostasis [66,67]. Taken together, many studies have shown the loss or inactivation of p53 causes centrosome amplification likely through multiple pathways, although the centrosomal targets and protein interactions of these processes still needs further research.

Another important tumor suppressor protein involved in CA is pRB. Much of our understanding of how pRB contributes to CA has come from studies on human papilloma virus (HPV)-encoded oncoprotein E7 [68]. Expression of HPV-E7 destabilizes pRB and interferes with the CDK inhibitor p21 [69]. Interestingly, expression of E7 leads to abnormal centrosome synthesis occurring prior to nuclear abnormalities developing, whereas expression of the E6 oncoprotein leads to the aberrant centrosome duplication occurring at the same time as nuclear abnormalities [68]. Given that expression of E6 primarily targets p53 for ubiquitin-dependent degradation, it is speculated that E6 supports the survival of the cells with mitotic disorders [65]. It has also been shown that acute loss of pRB can cause CA and concomitant chromosomal instability in murine primary fibroblasts and a similar induction of CA and aneuploidy is observed when RNAi-mediated knockdown is performed in human primary fibroblasts [70].

Several other tumor suppressor genes have been demonstrated to be involved in CA when misregulated. *BRCA1* is frequently mutated in cancers, and those mutations have been associated with CA [71]. BRCA1 upregulates p21 [72], suggesting that p21′s role in inducing CA may arise from the effects of *BRCA1* mutations in addition to, or in place of, the aforementioned p53-p21 axis. Additionally, PTEN redistributes from nuclear and cytoplasmic localizations to the centrosomes as cells enter mitosis, peaking in prophase and prometaphase and this localization was characterized to protect the integrity of mitotic centrosomes [73]. Inhibition of Akt prevented the recruitment of PTEN to centrosomes, and reduction of PTEN and Akt levels resulted in centrosomal defects, suggesting both are required for the proper regulation of mitotic centrosomes [73].

Other than the loss of function mutations of tumor suppressor genes, oncogenic induction of tumor cells also results in amplified centrosomes. MYC and Ras are well-characterized oncogenes and both of them were shown to contribute to CA. Overexpression of c-MYC was found to induce CA by inhibiting the negative regulator of PLK4, SCF/CUL1, while promoting activity of the positive regulator, Cyclin E/CDK2 [74]. The oligopeptidase tripeptidyl peptidase II (TPPII) was found to be involved in CA, being upregulated in Burkitt lymphoma cells overexpressing c-MYC [75,76]. Overexpression of TPPII disrupted centrosomes via centriole multiplication, a state where multiple procentrioles are nucleated from a single mother centriole, and a subsequent knockdown and/or inactivation of TPPII inhibited c-MYC-induced centrosome errors. Thus, TPPII is a necessary factor for c-MYC induced CA in tumors [77]. In addition to c-MYC, N-MYC was also shown to contribute to CA in a p53-dependent manner [78]. This study found that in order for N-MYC directed CA to occur, the cell requires MDM2-mediated suppression of p53 activity; furthermore, reactivation of p53 with Nutlin-3A was sufficient to completely inhibit N-MYC directed CA [78]. Interestingly, a study examining the contribution of Ras and c-MYC to breast cancer tumorigenesis compared breast cancer initiation by c-Myc and K-Ras (G12D) in early malignancy in vivo, and intriguingly found that while both could lead to CA in tumors, only K-Ras could induce CA during pre-malignancy. In this case, CA was associated with increased expression of Cyclin D1, CDK4, and Nek2 [79]. This again demonstrates that there are many pathways that lead to CA. It also suggests that some are more likely to be exploited at particular stages of tumorigenesis than others.

#### 3.1.2. Centrosomal Proteins

The Polo-like kinases are a family of five Ser/Thr kinases (PLK1-5) that have roles in cell cycle progression, the centrosome cycle, and mitosis. Centrosomal localization of all PLK isoforms have been reported in humans [80]. PLK4 is the keystone centrosomal duplication protein and increased expression of PLK4 has been reported in cancers originating from different tissues and organs including colon, stomach, breast, prostate, and brain [8,81,82,83,84,85]. By contrast, decreased PLK4 expression is observed in hepatocellular carcinoma [86] and hematological malignancies [87]. mRNA expression of PLK4 is regulated by p53 through recruitment of HDACs to the promoter region of the *PLK4* [88] and loss of p53 in tumors is discussed as a contributing factor to the increased PLK4 expression [89]. Tumors that developed after PLK4 induced CA were also correlated with reduced p53 expression [49]. In addition, expression of the HPV-16 E7 oncoprotein was found to increase PLK4 levels [90]. Different studies in p53-deficient mice has shown that overexpression of PLK4 can accelerate tumorigenesis in epidermis [91] as well as induce CA, tissue hyperplasia, and loss of primary cilia [92].

PLK4 activity is crucial for centriole biogenesis and excess PLK4 levels lead to production of extra centrioles around a mother centriole, forming a structure named a rosette (or flower) centrosome due to its unique shape [22,93] (Figure 2). The first observation of rosette centrosomes was made in 1971 using thin-section electron microscopy [94,95]. Later, Habedanck et al. (2005) reported the formation of ‘‘flower-like structures’’ in cells overexpressing PLK4 [21]. Kleylein-Sohn et al. (2007) showed multiple centrioles in rosettes form during S phase and persist throughout S and G2 phases, suggesting rosette centrosomes are functional as a whole structure and cycle like normal centrosomes [22]. Kuriyama (2009) showed simultaneous overexpression of PLK4, HsSAS6, and SAS4 in CHO cells resulted in the formation of rosette centrosomes [95]. In addition, Cosenza et al. (2017) observed the presence of rosette centrosomes in primary tumor samples including multiple myeloma, glioblastoma and colon cancer samples, highlighting that the generation of these structures is observable in naturally occurring tumors, and are not just an artifact of genetic or pharmacological manipulation of cells [93]. The study also identified that overexpression of STIL is capable of generating rosette centrosomes [93]. A recent paper from Ching et al. showed that centrioles in olfactory sensory neurons are amplified in precursor cells via formation of centriole rosette structures, suggesting rosette centrosome formation is also required for the generation of normal multiciliated cells [96].

Inhibition of PLK4 with small molecule inhibitors has been considered as a therapeutic option in cancers [97,98]. A reversible PLK4 inhibitor, centrinone, led to centrosome depletion in cells, but was found to be effective at inhibiting proliferation only in normal cells; treatment resulted in a senescent-like state that was dependent upon p53 [99]. However, this study found that cancer cells were capable of proliferating even after centrosome loss induced by centrinone treatment and that after washout of the inhibitor, the cancer cells reverted to a distribution of centrosome number similar to untreated cells over time. This suggests that centrosome loss affects normal cells and cancer cells in different ways. A second inhibitor, CFI-400945, described as a potent and selective PLK4 inhibitor, was found effective in cancers, and taken into clinical trials [100,101]. However, subsequent studies have suggested that there may be important off-target effects of CFI-400945 to be considered [102,103] and the debate continues with the suggestion that CFI-400945 may promote overduplication of centrosomes at low concentrations, but deplete centrosomes at higher concentrations due to the stability and activity of PLK4 at those concentrations of treatment [104]. Nevertheless, PLK4 and PLK4 induced CA still provide interesting targets for small molecule inhibitors. A recent paper identified the ubiquitin ligase TRIM37 as responsible for cancer-specific vulnerability to PLK4 inhibition [105], and identification of vulnerability hot spots could help PLK4 targeting in cancers.

Other than PLK4, other PLK isoforms have also been investigated broadly. An increase in PLK1 expression is well documented in numerous cancers [106,107,108,109]. PLK1 was shown to regulate Mst2-Nek2A induced centrosome disjunction by phosphorylating Mst2 [20]. PLK1 also regulates separase activity and centriole disengagement [110]. In addition, depletion of PLK1 expression resulted in inhibition of CA in hydroxyurea-treated centrosome amplified U2OS cells, suggesting overexpression of PLK1 in cancer contributes to CA [111]. However, PLK1 overexpression has also been shown to result in segregational and cytokinetic defects, thus generating polyploid cells; this prevents the development of KRAS- and HER2-induced mammary tumors in vivo, suggesting the contribution of intracellular PLK1 expression levels to carcinogenesis could be both dose- and context-dependent [112].

PLK2 and PLK3 have been proposed to act as tumor suppressor proteins [86,113], and the loss of PLK2 was identified as a common change found in colorectal carcinomas, lending support to PLK2 being identified as a tumor suppressor [114]. Additionally, silencing of PLK2 in gastric cancer cells results in increased proliferation and decreased apoptosis [115]. Contrariwise, other research found that PLK2 activity promotes tumor growth and inhibits apoptosis of colorectal cancer cells in vitro and in vivo [116]. This research found that binding of PLK2 to the tumor suppressor Fbxw7 resulted in increased degradation of Fbxw7 and stabilization of Cyclin E; this suggested that PLK2 may act as a tumor supportive factor, may serve as a prognostic and diagnostic target and furthermore as a therapeutic target [116]. Similarly, PLK3 was defined as a tumor suppressor, and under hypoxic conditions, functions as a negative regulator of HIF-1α [117]. In addition, other studies have demonstrated that overexpression of PLK3 results in shortened survival time of patients with hepatocellular carcinoma [118,119], and PLK3 is upregulated in breast and ovarian cancers [120]. However, the PLK3 expression level appears to be downregulated in induced colon tumors [121], as well as in bladder [119], uterus [119], and head and neck tumors [122], suggesting that the tumor suppressor function for PLK3 cannot be generalized, but it depends on the context of the individual tumor type [123]. That the upregulation and downregulation of PLK1, PLK2, and PLK3 are capable of inducing tumorigenesis under different circumstances in different tumors underlies the multifaceted roles these kinases play in normal cell cycle progression, and how misexpression of these proteins can lead to tumor progression. These seemingly contradictory findings from different studies demonstrate that the functions of these PLK family proteins in cancer cell biology still need further investigation.

In addition to the PLK family, other proteins whose importance in centrosome regulation and contribution to CA have been partially elucidated. One such protein is HsSAS-6, whose expression level is important for the restriction of procentriole formation: overexpression of HsSAS-6 promotes the formation of the previously mentioned rosette structures consisting of excess procentrioles [124]. Increased HsSAS-6 mRNA and protein expression levels were observed in human primary colorectal carcinomas, and led to CA, mitotic abnormalities and increased chromosomal instability [125]. This finding led to an analysis of expression levels of HsSAS-6 in various cancer types that were catalogued in publicly available TCGA RNA-Seq databases; the analysis found that HsSAS-6 is overexpressed in numerous cancer tissues when compared to normal non-cancerous tissues; tissues determined to show overexpressed HsSAS-6 included bladder, kidney, breast, lung, prostate and pancreas [125]. Similarly, defects with procentriole assembly and subsequent chromosome segregation occur when CEP152 is defective: the mutations E21K and V8A have been identified to impair the interaction of CEP152 with PLK4 [36].

In addition to CA, the deregulation of centriole size was identified as another centrosomal feature of cancer cells. Recent research indicates that centriole over-elongation results in enlarged centrosomes and these have increased microtubule nucleation capacities and promote chromosome missegregation [46]. Overexpression of CPAP, an important CE factor, was shown to result in the formation of abnormally long centrioles. This led to cells having supernumerary MTOCs that then subsequently led to multipolar spindles and cytokinesis defects [126]. As a further consequence, centriole over-elongation also induces CA via ectopic procentriole formation [40,46,127].

### 3.2. Indirect Centrosome Amplification

The discussed alterations in gene expression can be considered direct mechanisms that result in CA, but supernumerary centrosomes can also originate from indirect mechanisms. These mechanisms include: (i) cell-cell fusion; (ii) failure of cytokinesis; (iii) premature centriole disengagement; and (iv) PCM fragmentation. These methods increase CA by means other than increased centriole biogenesis or duplication. (i) When cell-cell fusion occurs, two cells with the normal centrosome complement fuse together, generating a single cell that possesses twice the amount of DNA and double the number of centrosomes. (ii) A failure of cytokinesis means that a cell that has undergone both DNA replication and centrosome duplication but does not complete cytokinesis towards the end of mitosis, and reverts back to a single cell with twice the DNA and centrosome number present. (iii) Premature centriole disengagement drives early separation of the daughter centriole from the mother, and results in a cell with a normal DNA content, but more than two centrosomes. Of these excess centrosomes, at least two will consist of only a single centriole. (iv) PCM fragmentation occurs when significant amounts of PCM components aggregate in random places in the cytoplasm.

To identify the presence, prevalence, and impact of these types of CA, researchers have used fluorescent imaging. Antibodies specific to centrioles (e.g., Centrin-2, Centrin-3) and PCM components (e.g., γ-tubulin) in fixed cells, and fluorescently tagged proteins in live cells have proved invaluable tools in identifying each type of CA [127]. This approach of using both live and fixed cells is necessary; for example, it is infeasible to use fixed cells to understand the differences between cell-cell fusion and failures of cytokinesis and thus live cell imaging is critical.

#### 3.2.1. Cell Fusion and Failures of Cytokinesis

Cell-cell fusions and cytokinesis failures result in cells that have twice the DNA and centrosome content. The fusion of two diploid (2n) cells generates a tetraploid (4n) hybrid cell, and often the nuclei of these fused cells remain separate from each other. However, sometimes the two nuclei will fuse and this can contribute to aneuploidy and cancer [128].

One key player in cell-cell fusion is RAD6. This protein, primarily known as an ubiquitin-conjugating enzyme, was found to be expressed and upregulated in metastatic breast tumors. The constitutive overexpression of RAD6 resulted in cell fusion, and subsequently, the generation of multinucleated cells, CA, multipolar mitotic spindles, and aneuploidy [129]. While this seems a straightforward association, cell fusion does not always cause aneuploidy and chromosomal instability, despite the fact that most of the fused cells have more than two centrosomes [130]. It is hypothesized that while CA caused by cell fusion can lead to multipolar spindle formation and therefore chromosomal instability in early stages of tumorigenesis, clonal outgrowth preferentially supports cells with a stable genome after tumor progression [128]. This could be achieved via centrosome clustering mechanisms, discussed in detail in later sections. Other than CA, cell fusion is also correlated with drug resistance and metastasis, serving a potential target for cancer therapy [131,132].

Cell division failure (failure of cytokinesis) can occur via many pathways. These pathways include the existence of a physical obstruction of the cleavage furrow, altered expression of regulators of cytokinesis, mutations in cytokinetic drivers, or mitotic slippage [133]. From these or other methods of failure of cytokinesis, the resulting cell will possess extra centrosomes as it re-enters interphase. As that cell progresses through the cell cycle, it will duplicate all of its centrosomes, ending with an excess number [134]. Failure of cytokinesis has significant tumorigenic potential: transient blockage of cytokinesis in p53^−/−^ mouse mammary epithelial cells generates tetraploid cells, and these cells exhibited an increased frequency of chromosomal alterations and in vivo tumorigenic potential compared to their diploid counterparts [135]. It is postulated that tetraploid cells generated by cytokinesis defects are better able to tolerate the loss of chromosomes, thus allowing them to produce more viable aneuploid progeny [133,136]. Failure of cytokinesis was also reported to promote aneuploidy via multipolar mitosis in glioblastoma cells [137]. There are some reports that question a direct correlation between failure of cytokinesis and CA: one study indicated that repeated cleavage failure did not establish CA in untransformed human cells, and that cytokinetic failure resulted in only a small increase in CA in p53^−/−^ HCT116 cells, but a relatively high increase in CHO cells. This suggests that the consequences of cleavage failure on increased CA frequency is likely cell type dependent [138].

A well-known and frequently observed characteristic of cancer cells is dicentric chromosomes. Chromosomes having two centromeres possess high potentiality to induce genome instability [139]. Centromeres of dicentric chromosomes tend to migrate towards the opposite poles during cell division, causing repetitive events of chromosomal breakages and re-ligations. Concomitant rearrangements of dicentric chromosomes occur due to the breakage-fusion-bridge (BFB) cycle [140,141]. Aberrations in gene copy numbers due to amplifications and deletions increases the likelihood of malignancy. Dicentric chromosomes are generated as a result of telomeric fusions [142] or form due to the presence of double strand breaks [143]. Repetitive DNA replications leads to telomere erosion and thus the formation of sticky ends on the chromosome. Chromosomal bridges generated by the fusions of telomeric regions of different chromosomes lead to cytokinetic failures, hence aneuploidy and cells bearing extra centrosomes [144,145,146]. Independent studies have discussed the correlation between centrosome amplification status and aneuploidy caused by telomere erosion. To clarify the origin of genome instability, researchers investigated centrosome amplification in different multiple myeloma stages, a cancer type well characterized by aneuploidy. They highlighted that centrosome amplification is frequently present even in early stages of myeloma. Moreover, they reported that there is no significant difference between ploidy subtypes (hyperdiploid: whole chromosome gains/losses; non-hyperdiploid: structural abnormalities and translocations) in terms of centrosome amplification [147]. This implied that telomere fusions and centrosome amplification contribute to CIN separately in the case of multiple myeloma. In another study, it was reported that telomere dysfunction and p16^INK4a^ deficiency in breast cancer cooperatively caused centrosomal aberrations in both diploid and polyploid cells, even in the presence of functional p53 [148]. A novel molecular mechanism between telomeres and centrosomes was recently identified by another group. Telomerase transcriptional element interacting factor (TEIF), primarily known for its function in hTERT activation, was shown to localize to centrosomes in all stages of the cell cycle, and centrosomal recruitment was increased by telomere dysfunction [149]. In addition, EGF/PI3K signaling was identified as an important regulator of centrosomal recruitment of TEIF, resulting in centrosome amplification [150]. In colorectal cancers, a positive correlation between TEIF expression and centrosome amplification was reported [151], indicating that TEIF could be playing a crucial role mediating telomere dysfunction and centrosome aberrations.

#### 3.2.2. Premature Centrosome Disengagement and PCM Fragmentation

While both processes involve the separation of parts of a centrosome, centriole disengagement is different from the centrosome separation. Centrosome separation occurs at the onset of mitosis as the duplicated centrosomes are separated to form the two spindle poles. As separation is initiated, centriolar linker proteins are phosphorylated, ensuring the connection between mother centrioles is severed by the onset of mitosis. This connection is made of linker proteins such as C-Nap1, rootletin, centlein, CEP68, and LRRC45 [152,153,154,155,156]. Initiation of separation occurs when Nek2A phosphorylates C-Nap1 and Rootletin [156,157,158,159,160] leading to an alteration of rootletin’s interaction with β-catenin [161]. After separation, centrosomes recruit additional PCM as they mature, thus allowing them to facilitate spindle assembly. Upon completion of mitosis, each daughter cell receives one centrosome, which consists of a mother-daughter pair that are tightly bound together. As the cell proceed through anaphase to next G1, the bound centriole pair disengage from each other, and after disengagement, the centrioles maintain connections to one another via the aforementioned centriolar linker proteins [18,162].

By comparison, premature centriole disengagement is not a normal phenomenon that occurs in healthy cells. When this process occurs, it will do so around metaphase and can result in the generation of multipolar spindles [127,163]. Centriole disengagement depends on the activity of separase, which is primarily known for its role in sister chromatid separation: separase hydrolyzes cohesin, which holds sister chromatids together [164]. However, separase also acts on centriole disengagement with numerous studies demonstrating an association between high separase activity and increased rates of multipolar spindle formation [165,166]. Separase functions by cleaving Scc1 (the cohesive keystone of the cohesin complex), and it has been observed that Scc1 is also localized to the centrosome, suggesting a role for Scc1 in maintaining centriole connections. Depletion of Scc1 also induced centriole splitting [167]. Furthermore, depletion of astrin, a microtubule and kinetochore protein, resulted in untimely separase activity and multipolar spindles, suggesting astrin contributes to the regulation of separase activity [166]. The Akt kinase interacting protein (Aki1) also seems to be involved in this process. Aki1 localizes to centrosomes and its depletion causes separase-dependent centriole splitting and multipolar spindles [167]. Separase’s ability to cleave both sister chromatids and centrioles was explored temporally with real-time imaging: using sensors, researchers could determine the point in mitosis at which separase’s activity was initiated. They found that centrosomal sensors were cleaved by separase before the onset of anaphase, and that this was earlier than they observed the chromosomal sensor being cleaved [110]. The activity of separase is inhibited by the spindle assembly checkpoint (SAC), thus mitotic delay or arrest may result in centriole disengagement. The SAC functions as a safety device to ensure proper, timely chromosome segregation and prevent missegregation [168]. SAC functions by delaying mitotic progression until the bipolar orientation of all chromatids has been achieved.

As SAC-induced mitotic delay persists, premature centriole disengagement and centrosome fragmentation follows; in order to maintain a bipolar spindle and division, the cell relies upon KIFC1(also known as HSET)-mediated centrosome clustering [163]. In addition, induction of mild replicative stress in non-cancerous cells was found to be sufficient to cause premature centriole disengagement, ultimately leading to transient multipolar spindles and lagging chromosomes. This early disengagement was related to the activity of CDK, PLK1, and ATR kinases during G2. This demonstrated that DNA damage can induce errors in centrosome cycle regulation [169].

Another mechanism by which premature centriole disengagement can occur was demonstrated by depleting the spindle-kinetochore proteins SKA3, CENP-E, or Cdc20. This led to the dysregulation of coordinated sister chromatid separation, a process termed cohesion fatigue; a consequence of cohesion fatigue was premature centriole disengagement and multipolar spindle formation [127,170]. Furthermore, recovery from pharmacological microtubule inhibition by colcemid or nocodazole leads to multipolar spindle formation with atypical centrioles at spindle poles. The poles formed here can be mono-centriolar or acentriolar in nature [128,159,160]. Additionally, nitrous oxide treatment [171] and heat shock [172] produced similar outcomes.

Independent from premature centriole disengagement, accumulating knowledge shows that PCM fragmentation is commonly observed in cancers. PCM fragmentation is defined as the generation of acentriolar PCM in random locations, from which microtubules can be nucleated and they are capable of functioning as spindle poles. One study examining CA in breast cancer found that high-grade tumors possess greater numbers of acentriolar centrosomes, suggesting PCM fragmentation may be common in breast cancers, and the rate of PCM fragmentation could increase with tumor stage and grade [173].

Research about the molecular mechanisms of PCM fragmentation has started to get attention, and several proteins have been identified as important contributing factors to centrosome integrity. Inactivation of Aurora A kinase, via either siRNA-mediated knockdown or a specific chemical inhibitor, is sufficient to cause PCM fragmentation and microtubule hyperstabilization [174]. Another key protein is kizuna, a centrosomal protein that is phosphorylated by PLK1 during mitosis and is critical for centrosome integrity and stabilization of the PCM [175,176]. Reduced expression of kizuna was shown to result in centrosome fragmentation and dispersion of PCM, leading to multipolarity and chromosome segregation errors [177]. This suggested that proper regulation and function of kizuna is a fundamental step of maintaining structural integrity of the centrosomes as they are exposed to the forces necessary to generate spindles [62]. In a recent study, a bioinformatic approach was used to identify novel mitotic components, and subsequent analysis revealed the role of chondrosarcoma-associated gene 1 protein (CSAG1) to be involved in maintenance of centrosome integrity during mitosis. Depletion of CSAG1 led to acentriolar multipolar spindle formation; this was especially pronounced in p53-compromised cells [178]. Moreover, siRNA-mediated knockdown of CEP164 was also demonstrated to result in the formation of multipolar spindles with acentriolar poles, identifying another protein that induces PCM fragmentation [179].

Pericentriolar satellites are granules that are found around the centrosome [180,181] and are involved in the recruitment of centrosomal components as well as modulation of microtubule organization [182]. Proper maintenance of these sites appears critical to preventing PCM fragmentation. Members of the CLASP (cytoplasmic linker-associated protein) family are known to be regulators of polarity in cell division, and alterations in CLASP expression leads to dysregulation of bipolar spindles [183]. CLASP1 regulates kinetochore-microtubule dynamics along with astrin and Kif2b [184]. However, a study found that siRNA-mediated knockdown of CLASP1/2 caused the generation of acentriolar spindle poles, suggesting the loss of these proteins could lead to PCM fragmentation. Further analysis revealed that CLASPs ensure spindle-pole integrity by recruiting ninein to pericentriolar satellites [185]. Additionally, CEP90 is characterized as a pericentriolar satellite localizing protein, and depletion of CEP90 causes spindle pole fragmentation and mitotic arrest, suggesting CEP90′s role at pericentriolar satellites is crucial for maintaining the integrity of spindle poles during mitosis [186]. Taken together, these studies demonstrate that pericentriolar satellites are vital for maintaining centrosomal integrity as additional forces are applied to the structures as the spindle is assembled.

## 4. Outcomes of Centrosome Amplification

In spite the deleterious effects extra centrosomes have on cell viability, the phenotype persists in cancer cells. This raises the question of why this is so, and suggests that CA offers an advantage to these cells or at least is able to turn this negative situation into a beneficial one. The advantages to a surviving cancer cell include chromosomal instability, alterations to cell polarity, motility, and intracellular signaling. All of these can be positive factors for cancer cells, providing them with survival and proliferative advantages. For example, genomic instability caused by CA can increase tumor heterogeneity, thus promoting the possibility of better compatibility with the microenvironment [187]. Thus, maintaining supernumerary centrosomes could be a survival and fitness mechanism for cancer cells; the dividends of CA-driven alterations outweigh the disadvantages associated with having extra centrosomes.

Examinations of cancer cells, both in situ and in culture, reveal that many exhibit extra centrosomes, and the presence of these have been claimed to contribute to both genomic instability as well as tumor progression. Genomic instability, a hallmark of cancer progression, mainly stems from chromosomal instability (CIN); with microsatellite instability (MIN) proposed as a lesser trigger of CIN. CIN is the rate of change in chromosome number or structure and is tied to aneuploidy, a description of the state of change of chromosomes from the normal ploidy of the cell. The mechanisms by which CIN occurs and leads to aneuploidy remain elusive, though several causes have been implicated. Along with disruption of mitotic checkpoints, incomplete chromosome condensation and defective microtubule-kinetochore attachments, CA has been suggested as a significant contributor to CIN [188]. It remains unclear if CA directly leads to tumorigenesis, or it manifests as a byproduct of other cell cycle defects. Numerous studies have demonstrated that CA is abundant in a variety of tumors including breast, prostate, ovarian, head, neck, lung, and bone, and that its prevalence occurs alongside increased rates of CIN and aneuploidy. This suggests that CA indeed may be a major contributor to tumorigenesis [8]. Furthermore, the centrosome cycle involves numerous oncoproteins and tumor suppressor proteins, underlining that cancer may itself induce abnormalities in centrosome number and function, thus making CA and tumor progression occur hand-in-hand [62].

While aneuploidy and CA frequently co-exist in cancer cells [189], having supernumerary centrosomes would appear to be an obstacle to the cell. Extra centrosomes can result in multipolar divisions, likely to lead to the loss of genes essential for the survival of a diploid cell. However, cancer cells are frequently found to be tetraploid, and multipolar division under those circumstances is much more likely to generate not only viable progeny, but also progeny that have advantageous chromosomal rearrangements. [65]. In diploid cells, CA-induced multipolarity will lead to non-viability, but clustering of the extra centrosomes into a bipolar spindle allows proliferation to continue. These clustered spindles maintain the proliferative cycle longer while also serving as the primary mechanism by which CIN and aneuploidy can be induced [190].

Centrosome defects can be seen in most aggressive carcinomas: both CIN and cell cycle errors induced by supernumerary centrosomes can contribute to robust tumor development. CA in the early stages of cancer can accelerate CIN, resulting in the accumulation of oncogenic mutations and the loss of tumor suppressors. Pihan et al. demonstrated that centrosome abnormalities might drive the evolution of pre-invasive and early stage lesions into aggressive and high pathological grade tumors in cervix, breast, and prostate tissues [45]. Likewise, D’Assoro et al. proposed CA as a prognostic marker for breast tumors exhibiting genomic instability, indicating aggressiveness [191]. Similar results were obtained for bladder cancer and chronic myeloid leukemia, indicating that CA-dependent genomic instability promotes tumor progression, elevates the pathological stage of the tumor, and predisposes tumor recurrence [192,193].

Further evidence of CA’s role in tumorigenesis was demonstrated using mouse models [91]. Here, PLK4 overexpression was used to induce CA during epidermal development. CA led to the stabilization of p53 and subsequently induced apoptosis and skin defects. When PLK4 overexpression was combined with p53 knockout, mice were prone to develop skin cancer. An additional study using transgenic mice overexpressing PLK4, inducing CA, showed that tumorigenesis was enhanced in the absence of p53: cells in pancreas and skin tissue exhibited accelerated proliferation that correlated with an increase in centrosome number [92]. Overall, this demonstrated that CA was a strong contributor to tumor progression and this activity is greatly enhanced by the absence of p53 activity.

While clustering of extra centrosomes to form only two poles is favorable for cell viability, abnormal attachments of chromosomes and microtubules are prevalent in these cells. Cancer cells with extra centrosomes can form impermanent multipolar spindle assemblies, which later proceed to a pseudo-bipolar state by centrosome clustering [65]. The clustering process results in the generation of merotelic kinetochore attachments [194], which are not detected by the spindle assembly checkpoint (SAC) and which produce segregational defects. Most commonly, a merotelic attachment that evades the SAC produces lagging chromosomes [195], a significant source of aneuploidy and CIN [196]. In addition, lagging chromosomes form micronuclei, which contribute to both DNA replication errors and DNA damage in daughter cells [196,197,198]. As is well established, an increase in DNA damage leads to more mutagenesis likely to drive tumorigenesis. Thus, clustering of centrosomes after CA will drive tumor cells towards an aneuploid state. Therefore, CIN is a double-edged sword for cancer cells: it can either promote proliferation or act as a suppressor of growth. The CIN-dependent increase in the rate of tumorigenesis is likely due to an elevated rate of mutations in subpopulations of the emerging cancer cells [199]. This state of genomic heterogeneity caused by CIN provides the opportunity for cancer cells to acquire emerging traits that escalate the chance of survival [187]. The degree of CIN is, therefore, the main determinant of survivability for cancer cells. At lower levels, genetic instability is tolerated, even favorable, and advantageous for tumor progression; however, higher levels of CIN are detrimental and potentially lethal to cancer cells [200]. Overall, CIN may act as a significant selective stress for cancer cells to keep their extra centrosomes.

In addition to contributing to tumorigenesis by driving CIN, CA can induce tumorigenesis by impeding symmetrical cell division. One study grafted neural stem cells containing amplified centrosomes to Drosophila; tumors that formed in these organisms led to rapid death, even though the tumor samples exhibited low levels of aneuploidy [48]. This suggested a mechanism other than aneuploidy or CIN was driving tumor formation, and that this occurs despite efficient clustering of centrosomes during mitosis. Although the cell is capable of clustering extra centrosomes to undergo bipolar division, the resulting daughter cells are generally not identical in terms of both genomic and cytoplasmic content. Thus, efficient clustering of centrosomes still disrupts the polarity of dividing cells, and this asymmetric division of stem cells may generate subpopulations that are more efficient at differentiating into tumor cells. This is in contrast to when centrosome clustering efficiency is low: those cells tend to undergo multipolar division and the progeny exhibit high levels of severe aneuploidy. One study observed that this state selects against tumor formation, demonstrating that clustering of centrosomes is a necessary step in maintaining cells in a proliferative state as tumors form [201].

As well as their key role in mitosis, centrosomes are also responsible for microtubule organization, which is vital for cell signaling, cell polarity, morphology, and motility [202]. Therefore, amplification of centrosomes can lead to alterations to cellular cytoarchitecture, thus leading to changes in the organization of the tumor tissue and the behavior of the cells within. In this context, centrosome amplification could be a contributory factor to metastasis. In breast cancer, it was reported that cells with supernumerary centrosomes and high clustering ability promoted high grades of tumorigenesis and metastasis [203]. The study underlines that CA-associated tumor aggressiveness is not necessarily CIN dependent.

During interphase, clustered extra centrosomes can recruit excess PCM, forming a giant centrosome. This structure can nucleate more microtubules, leading to an aberrant microtubule array, and altered cell polarity. These disorganized microtubules can deleteriously affect cell morphology and motility. Furthermore, correct positioning of centrosomes is necessary for the establishment of cell polarity, a critical process in development [204]. Enlarged microtubule organizing centers lead to greater polarization, though the link to cancer biology has not been established. Cell motility is also directed in part by microtubules, where the activity of Rho GTPases are activated by guanine nucleotide exchange factors (GEFs), including p190Rho-GEF and GEF-H1 [205,206]. These monomeric GTPases regulate cell-cell adhesion and subsequent locomotive events, which are major steps in tumor progression [207]. It has been demonstrated that extra centrosomes can induce invasion in a three-dimensional culture model via the induction of the Rho GTPase Rac1 [208]. These data suggest that centrosome amplification leads to alterations in the cytoskeleton and that this change in organization and abundance of microtubules and associated factors promotes invasion and metastasis, thus serving as a pivotal step along the pathway of tumor malignancy.

Centrosomes are also involved in cell signal regulation, most prevalently during mitotic entry. Here, the centrosome serves as a site of regulation for several mitotic regulatory factors including cyclin B1/Cdk1 and the Aurora A kinase AIR-1, both of which are critical for timely mitotic entry [209]. Proteomic tools have allowed researchers to identify components of numerous cellular signaling pathways that localize to centrosomes [210,211,212]. These proteins include a number of oncogenic proteins involved in proliferative pathways, such as the integrin, Wnt and NF-κB pathways. Integrin-linked kinase (ILK) has been demonstrated to localize to mitotic centrosomes and interact with several centrosomal proteins including ch-TOG, RUVBL1, and α- and β-tubulin. In addition to functioning as an essential component of the integrin signaling pathway, regulating cell adhesion and migration, ILK activity is essential for mitotic spindle organization, functioning as a regulator of Aurora A/ch-TOG/TACC3 complex formation and thus spindle pole integrity [213,214]. For the Wnt signaling pathway, generally involved in modulating cell polarity and motility, it has been reported that pathway components must be localized to specific subcellular locations for its proper function. The inhibitory protein Diversin must be localized to the centrosome in order to effectively antagonize its specific targets in Wnt pathways [215]. Furthermore, Tax, a regulator of the NF-κB pathway is also localized to the centrosome. Tax is defined as an oncoprotein that promotes proliferation; it activates NF-κB via its interaction with IκB kinase (IKK) [216]. These three pathways serve as key examples of specific centrosomal components being pivotal to inducing tumorigenesis.

Since the centrosome is the site of accumulation for numerous proteins critical to signaling pathways that can be dysregulated, it follows that alterations in centrosome number or size can compromise the balance and dose of activity. Therefore, amplified centrosomes may contain excess amounts of signaling proteins, which may induce aberrant signaling leading to uncontrolled cellular proliferation and/or migration. On the other hand, CA also promotes stabilization of p53 and expression of p21. Activation of p21 induces cell cycle arrest and inhibits proliferation. Cells lacking p53 activity can circumvent this CA-induced cell cycle arrest and prosper in the presence of extra centrosomes [217].

## 5. Centrosome Clustering and Targeting of Clustering Mechanisms

Centrosome clustering is considered the primary mechanism by which cells with extra centrosomes prevent multipolar divisions, in both normal and tumor tissue. As such, understanding the mechanism by which clustering occurs is of significant interest. Centrosome clustering was first observed in interphase [218] and mitotic [219] N1E-115 mouse neuroblastoma cells, which possess exceptional levels of extra centrosomes. In several review articles, clustering was proposed as a potential option for targeted cancer therapy on tumors with extra centrosomes [47,220,221,222,223]: if clustering could be inhibited by small molecules, the cells with extra centrosomes would divide in a multipolar fashion; this increases the likelihood of cell death and decreases the rate of tumorigenesis (Figure 3). To achieve this goal, more information about the specific mechanisms of clustering must be uncovered.

One of the pioneer studies on the mechanisms for coalescence demonstrated that the microtubule motor cytoplasmic dynein played an important role in clustering; overexpression of spindle proteins, such as NuMA, resulted in a mislocalization of dynein, resulting in multipolar spindle formation [224]. Two genome-wide screens were executed to find additional target proteins that are responsible for centrosome clustering. These screens were performed on Drosophila [225] and human tumor tissue [179]. The latter study, an RNAi screen, compared multipolar spindle formation rates between cancer cells and normal fibroblasts, and prioritized targets specific for cancer cells. These studies spurred research to identify specific clustering factors, dividing them into three major classes: (i) cell division control and spindle tension proteins; (ii) spindle pole structure and centrosomal proteins; and (iii) actin organization and cell adhesion proteins. A summary of clustering mechanisms is given in Figure 4.

The first group identified in the screens consists of cell division proteins, including SAC components such as the chromosomal passenger complex, chromatid cohesion factors, kinetochore components, and spindle tension proteins such as the augmin complex [179]. The chromosomal passenger complex, formed by Aurora B kinase, INCENP, Survivin, and Borealin is sometimes referred to as the master regulator of cell division [226]. The augmin complex, consisting of eight subunits, plays a critical role in spindle microtubule generation via its interaction with the γ-tubulin ring complex (γ-TuRC) [227]. Screens have identified that depletion of any member of the chromosomal passenger complex leads to an increase in the rate of multipolar metaphases. Depletions of the augmin complex proteins FAM29A, HEI-C, and HAUS3 have similarly reduced clustering ability [179]. Disruption of spindle tension by mechanisms, such as reduced chromatid cohesion or misattachment of kinetochores was also sufficient to inhibit centrosome clustering. This was demonstrated by knockdown of many proteins, including HEC1, SPC24, SPC25, CENPT, sororin, and shugoshin (SGOL1) [179,225].

In addition, hepatoma upregulated protein (HURP) is a tension-sensitive kinetochore-stabilizing factor [228], which was recently identified as another required factor for centrosome clustering [229].

The second group of proteins required for centrosomal clustering is comprised of structural spindle pole and centrosome proteins. The principal hit in this group was the C-terminal kinesin motor KIFC1, which was first identified in the Drosophila screen [225]. Although KIFC1 was not a hit in the human cancer screen, many subsequent studies have confirmed its contribution to centrosome clustering in human tumor cells [90,220,221]. KIFC1 is a kinesin that binds to the minus ends of microtubules and crosslinks neighboring microtubules in the vicinity of the spindle pole [230]. Importantly, KIFC1 was also identified as a direct binding partner of CEP215, demonstrating its vital role in binding microtubules to centrosomes [231]. The CEP215-KIFC1 complex also promotes the clustering of extra centrosomes [47,231]. In addition, KIFC1 has been identified as a malignant cell dependency protein in breast cancers [232]. Interestingly, ATM and ATR phosphorylate KIFC1 upon DNA damage, in turn triggering centrosome clustering that leads to drug resistance and eventually to tumor recurrence [233]. Inhibition of this phosphorylation reverted both phenotypes, therefore there is ample evidence suggesting that the targeted inhibition of KIFC1 could be a treatment option in cancers with amplified centrosomes. Three KIFC1-specific inhibitors have been recently described: AZ82 [234], CW069 [235], and SR31527 [236]. AZ82 binds specifically to KIFC1, and treatment leads to centrosome declustering in BT-549 breast cancer cells that exhibit amplified centrosomes [234]. CW069 induces multipolarity only in cells that possess extra centrosomes [235] and SR31527 prevents centrosome clustering in triple negative breast cancer cells. Additionally, SR315627 reduces colony formation and cell viability in the same cells [236]. All three inhibitors are highly selective for cells with extra centrosomes when compared to cells with a normal centrosome complement. This suggests that KIFC1 functions as a specific target for centrosome declustering. In addition to its function in centrosome clustering, KIFC1 has also been shown to fuel tumor progression via centrosome-independent activities [237].

Several other centrosome and spindle pole structural proteins have been identified as vital to centrosome clustering. The Raf-like kinase Integrin-linked kinase (ILK), which functions in actin and mitotic microtubule organization, was identified as a clustering factor in several cell lines from breast and prostate origin that exhibited extra centrosomes [238]. ILK contributes to centrosome clustering through regulation of the microtubule associated proteins TACC3 and chTOG [213,239]. Perturbation of ILK led cancer cells to undergo multipolar anaphases, mitotic arrest, and cell death. Additionally, the study identified that ILK regulates Aurora A-dependent phosphorylation of TACC3, a necessary step for centrosome clustering [238].

CP110, a centrosome duplication regulatory factor, also contributes to centrosome clustering. Phosphorylation of CP110 by CDK2 was shown as a requirement for prevention of multipolar spindle formation [240]. Furthermore, depletion of the centrosomal protein CEP164 causes spindle pole disintegration and leads to generation of acentrosomal spindle poles [179]. Another study described the role of Nek6-dependent phosphorylation of Hsp72 in centrosome clustering [241]. Previous work from this group demonstrated that Nek6 activity was required for targeting Hsp72 to the centrosome, where it was responsible for kinetochore fiber assembly via recruitment of TACC3 and chTOG [213,242]. Depletion of Hsp72 increased multipolarity, but not via the formation of acentrosomal poles. Therefore, multipolar spindles were observed in cancer cells that had extra centrosomes, but loss of either Hsp72 or Nek6 function in normal cells did not affect spindle formation [241].

The final group of centrosome clustering proteins identified in the large-scale screens is proteins that function in linking actin and microtubules. Two key identified proteins here were Myo10 (otherwise known as myosin X) and cofilin [225]. Myo10, an N-terminal myosin, binds to microtubules and regulates spindle pole orientation [243]. Myo10 also plays a role in binding microtubules to centrosomes, facilitating proper orientation of centrosomes towards retraction fibers [243]. As observed with KIFC1 and others, depletion of Myo10 increased the frequency of multipolarity in cells with extra centrosomes, specifically S2 and N1E-115 cells, demonstrating that it also plays a role in centrosome clustering [225]. The actin-depolymerizing factor (ADF)/cofilin family is a three-member family consisting of small proteins that regulate actin dynamics by inducing filament severing [244]. A drug screen performed to identify centrosome-clustering inhibitors found two drugs, CP-673451 and crenolanib, which induced multipolar spindle formation via the cofilin-activated destabilization of the actin network. This established not only two potential tumor-targeting compounds but also the importance of cortical actin destabilization as a mechanism for inhibiting centrosome clustering [245]. The importance of cortical actin, specifically cortical contractility, is also demonstrated in epithelial cells, which express high levels of E-cadherin. These cells have low cortical contractility, and this is associated with reduced ability to cluster amplified centrosomes [246]. As cells undergo epithelial-to-mesenchymal transition, E-cadherin expression levels are reduced and there is a concurrent increase in cortical contractility that will allow KIFC1-dependent centrosome clustering [246].

Several outlying proteins that do not belong to the three previously described groups were also identified as important factors for centrosomal clustering. One such protein family is the deubiquitylase (DUB) enzymes, responsible for the removal of ubiquitin from proteins. The Drosophila siRNA screen identified two DUBs, USP8, and USP31, as important for centrosome clustering [225]. An earlier siRNA screen on human cancer cells also identified USP54 as another protein required for clustering of extra centrosomes [247]. Furthermore, other DUBs are required for different stages of the centrosome cycle: USP1 and USP33 are required for centrosome duplication, and USP44 is required for centrosome separation [248]. This suggests that the DUB family is involved in a complex regulatory network of centrosome duplication, regulatory, and clustering mechanisms, an area ripe for continued research to develop a better understanding.

Alongside DUBs, members of the E3 ubiquitin ligase TRIM family also contribute to centrosome amplification and clustering [249]. TRIM28, an ARF binding protein and an E3 SUMO ligase, SUMOylates NPM1, increasing its centrosomal localization and thus prevents CA [250]. TRIM19 represses Aurora A kinase activity, thus preventing CA [251]. Cells depleted for TRIM69A, a spindle pole-associated factor essential for mitotic spindle formation, exhibited mitotic defects. Additionally, it was found that TRIM69A overexpression could reverse HAUS1-induced spindle multipolarity, revealing TRIM69A as a potent clustering agent [252]. To add to this, RNAi-mediated depletion of TRIM69A inhibited tumor growth in vivo [252]. Lastly, *TRIM22*, an IFN inducible gene, also localizes to centrosomes independently from the cell cycle and promotes centrosome clustering [249,253].

In addition to the two siRNA screens, two drug-based screens have been performed to identify centrosome-clustering targets. The first of these used fungal extract libraries and identified griseofulvin as a potential hit [254]. Griseofulvin stabilizes microtubules at low concentrations, and studies revealed that the microtubule-binding site for griseofulvin overlaps with the paclitaxel binding site, suggesting a common mechanism by which multipolarity could be induced [89,255]. A synthetic derivative of griseofulvin, GF-15, was found to function as a potent inhibitor of centrosome clustering in malignant cells, and showed antitumor efficacy both in vitro and in vivo [256]. The second drug screen resulted in the identification of fourteen compounds that prevented clustering of centrosomes in BT-549 cells, a breast cancer line with extra centrosomes. Mitotic arrest was also observed with these 14 drugs. Of the 14, CCCI-01 was of particular interest, demonstrating a high differential response between the cancer cells and noncancerous mammary epithelial cells: while the BT-549 cells exhibited a high rate of multipolarity and cell death, the normal cells presented neither phenotype, marking CCCI-01 as a good candidate anti-cancer drug that targets the centrosomal clustering process [257].

Finally, a p53 abnormality was observed to impair centrosome clustering in tetraploid cells by modulating the RhoA/ROCK signaling pathway. This suggested that functional p53 was a requirement for centrosomal clustering, and it is needed to produce viable progeny from cells with extra centrosomes [258]. These data, derived from N/TERT1, 3T3, and mouse embryonic fibroblast lines, suggested that the mechanisms for clustering found in normal cells with functional p53 could differ considerably from those found in tumor cells lacking p53 activity. Given that tetraploid cell generation is required in some tissues (e.g., hepatocytes) [259,260], and that proper bipolar division in those cells relies upon centrosome clustering, better understanding of the mechanistic differences between centrosome clustering in cancer cells versus normal cells is crucial to determine. This is especially true as we discover more potential inhibitors of centrosome clustering, where we will desire drugs that target tumors while leaving normal tissues unaffected.

## 6. Conclusions

There has been a long-standing relationship between centrosome amplification and cancer progression. Multipolar divisions are detrimental for cells; therefore, cancer cells evolve molecular mechanisms to survive through clustering extra centrosomes and forming bipolar spindle poles during metaphase. Centrosome clustering is an essential requirement for cancer cells with supernumerary centrosomes. Nonetheless, it is still elusive as to which mechanisms are the most prevailing and whether it is a matter of life or death for cancer cells to manage adaptation against perpetual centrosome clustering events. Centrosome amplification is one of the predominant factors to induce CIN and accelerate tumor progression. As normal cells do not exhibit these anomalies, targeting cells with supernumerary centrosomes is an appealing idea to selectively kill cancer cells. Moreover, implementing novel strategies, successfully eradicating the cells bearing extra centrosomes within tumors, would exhibit a positive impact in the clinic. In conclusion, understanding the molecular mechanisms through which centrosomes are amplified, clustered, or inactivated, and how these events are linked to cancer progression, may lead to new therapeutic opportunities.

## Figures and Tables

**Figure 1 cancers-14-00442-f001:**
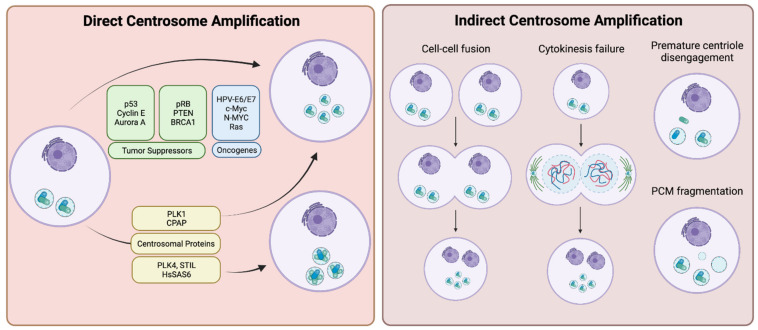
Centrosome amplification may occur directly through molecular changes within the cell such as overexpression of proteins regulating the centrosome cycle, presence of viral oncogenes, etc. (**left panel**), or indirectly via gross cellular anomalies, such as cellular fusion or premature centriole disengagement (**right panel**). Created with BioRender.com.

**Figure 2 cancers-14-00442-f002:**
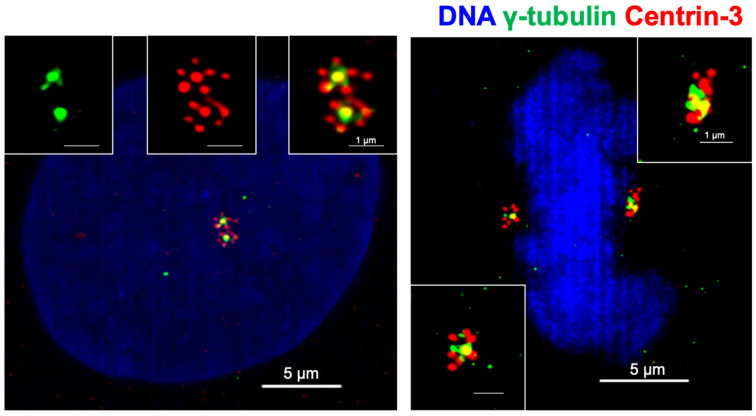
Centrioles in cancer cells form rosette like structures upon overexpression of PLK4. Rosette centrosomes can be readily seen in both interphase cells (**left**) and in mitosis (**right**). Blue: DNA (DAPI); green: centrosomes (γ-tubulin); red: centrioles (Centrin-3).

**Figure 3 cancers-14-00442-f003:**
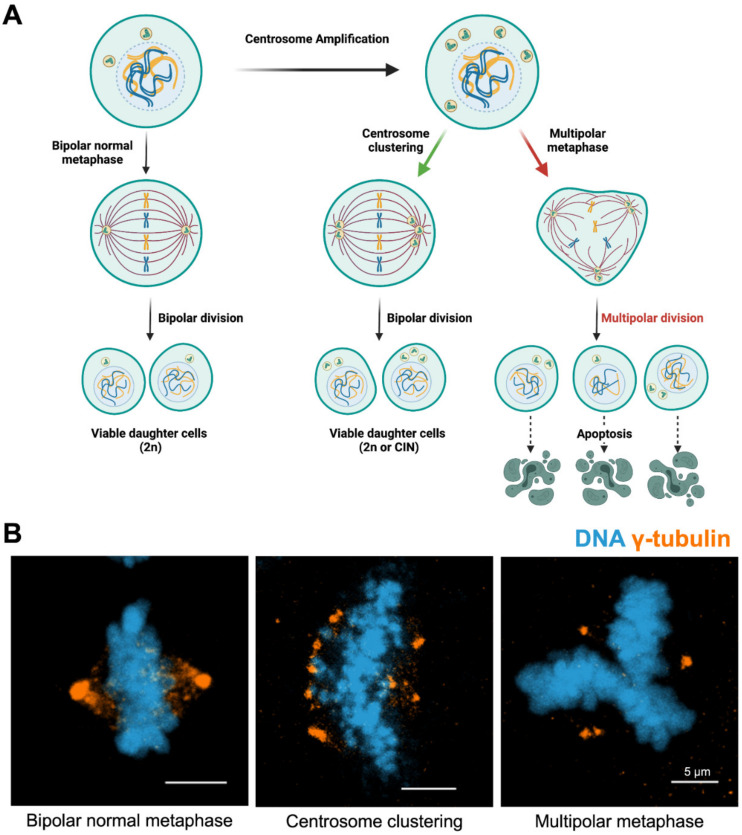
(**A**) Normal cells can effectively divide into two as they exhibit only two centrosomes, one at each pole. Extra centrosomes are capable of forming additional poles leading to multipolar divisions, which may result in loss of essential genetic material and eventually trigger cell death pathways. On the other hand, cancer cells can escape this fate through clustering their extra centrosomes and manage to divide in a bipolar fashion. Created with BioRender.com. (**B**) Confocal microscopy images displaying cells in metaphase: a normal bipolar division with one centrosome at each pole (**left**), a bipolar division with clustered centrosomes (**middle**), and a multipolar division with one or more centrosomes at each pole (**right**) are seen in the micrographs. Cyan: DNA (DAPI); orange: centrosomes (γ-tubulin).

**Figure 4 cancers-14-00442-f004:**
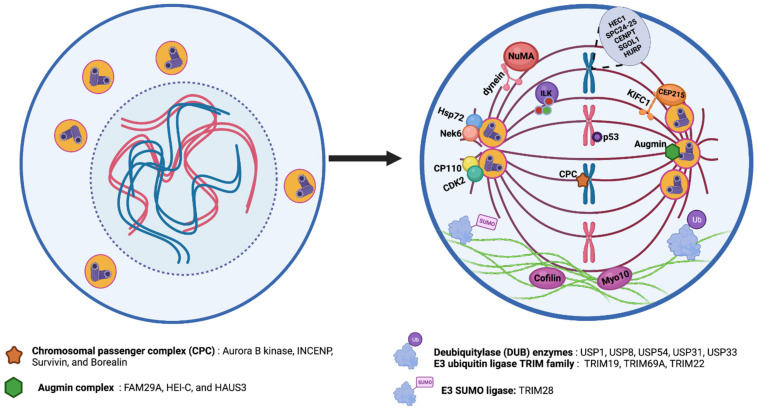
A summary of mechanisms leading to centrosomal clustering. Created with BioRender.com.

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
