# Peer review of "Keep Calm and Carry on with Extra Centrosomes"

_cancers, 2022, doi:10.3390/cancers14020442_

Round 1

Reviewer 1 Report

There is very little to comment on, other than to say that this is an informative and easily readable review of a subject that is not often considered in the common thinking about the role of aneuploidy in cancer, and, in particular, the potential of using centrosome biology in cancer diagnosis and treatment.  I look forward to its publication in Cancers. I have one comment in the legend of Figure 2 for clarity, suggesting it should read "Figure 2. Centrioles in cancer cells form..."

The English usage is very good and the text reads well. It feels like this ms. has been reviewed extensively before I saw it.  I suggest an additional detailed proofreading before publication, but I did not find any typos in my reading. 

Author Response

We thank our reviewer for these nice comments.

Reviewer 1:

There is very little to comment on, other than to say that this is an informative and easily readable review of a subject that is not often considered in the common thinking about the role of aneuploidy in cancer, and, in particular, the potential of using centrosome biology in cancer diagnosis and treatment.  I look forward to its publication in Cancers. I have one comment in the legend of Figure 2 for clarity, suggesting it should read "Figure 2. Centrioles in cancer cells form..."

The English usage is very good and the text reads well. It feels like this ms. has been reviewed extensively before I saw it.  I suggest an additional detailed proofreading before publication, but I did not find any typos in my reading. 

Response to Reviewer 1:

We changed the legend of Figure 2 as suggested by our reviewer. We thank the reviewer for these nice comments.

Reviewer 2 Report

Mert Kalkan   et al. describe the impact of centrosome amplification as a driving force of chromosomal instability in cancer progression. The manuscript covers several aspects of the subject and is well-organized. The paper is an update on the topic.

Nevertheless, some chapter is too long and need to be reduced (Outcomes of centrosome amplification, centrosome clustering and targeting of cluster …). The conclusion is very short and merits additional information such as the possible use of centrosome amplification as a methods to detect chromosomal instability.

The relationship between the centrosome amplification and the formation of dicentric chromosomes is missed as well as the correlation between centrosome amplification and telomere status. The interaction between the formation of specific configuration of dicentric chromosome related to the centromere breakpoints and telomere dysfunction could progress our knowledge of centrosome amplification and should be discussed in the manuscript.  

Author Response

We thank our reviewer for the comments. We beleive they helped us improve the manuscript.

Reviewer 2:

Mert Kalkan et al. describe the impact of centrosome amplification as a driving force of chromosomal instability in cancer progression. The manuscript covers several aspects of the subject and is well-organized. The paper is an update on the topic.

Nevertheless, some chapter is too long and need to be reduced (Outcomes of centrosome amplification, centrosome clustering and targeting of cluster …). The conclusion is very short and merits additional information such as the possible use of centrosome amplification as a methods to detect chromosomal instability.

The relationship between the centrosome amplification and the formation of dicentric chromosomes is missed as well as the correlation between centrosome amplification and telomere status. The interaction between the formation of specific configuration of dicentric chromosome related to the centromere breakpoints and telomere dysfunction could progress our knowledge of centrosome amplification and should be discussed in the manuscript.  

Response to Reviewer 2:

1- We revised the sections 4. and 5. that were mentioned by our reviewer and reduced the word count in both sections, albeit not dramatically. We could not cut the content for brevity as the affected proteins discussed in these sections are not more important or less important than the others. If our reviewer gives more specific examples about what, exactly, s/he thought was excessive, we are happy to do further editing. Having said that, we did clean up the text as much as we could to reduce the word count.

2- The conclusion section is lengthened to include additional points on use of centrosome amplification as an alternative mean of therapeutic option as suggested by our reviewer. The conclusion currently reads as:

“There has been a long-standing relationship between centrosome amplification and cancer progression. Multipolar divisions are detrimental for cells; therefore, cancer cells evolve molecular mechanisms to survive through clustering extra centrosomes and forming bipolar spindle poles during metaphase. Centrosome clustering is an essential requirement for cancer cells with supernumerary centrosomes. Nonetheless, it is still elusive that which mechanisms are the most prevailing and whether it is a matter of life or death for cancer cells to manage adaptation against perpetual centrosome clustering events. Centrosome amplification is one of the predominant factors to induce CIN and accelerate tumor progression. As normal cells do not exhibit these anomalies, targeting cells with supernumerary centrosomes is an appealing idea to selectively kill cancer cells. Moreover, implementing novel strategies successfully eradicating the cells bearing extra centrosomes within tumors would exhibit a positive impact in the clinic. In con-clusion, understanding the molecular mechanisms through which centrosomes are amplified, clustered or inactivated, and how these events are linked to cancer progres-sion may lead to new therapeutic opportunities.”

3- We added a new paragraph on the link between centrosome amplification, dicentric chromosomes and telomere dysfunction. The new information is pasted below.

“A well-known and frequently observed characteristic of cancer cells is dicentric chromosomes. Chromosomes having two centromeres possesses high potentiality to induce genome instability [139]. Centromeres of dicentric chromosomes tend to migrate towards the opposite poles during cell division, causing repetitive events of chromosomal breakages and re-ligations. Concomitant rearrangements of dicentric chromosomes occur due to the breakage-fusion-bridge (BFB) cycle [140,141]. Aberrations in gene copy numbers due to amplifications and deletions increases the likelihood of malignancy. Dicentric chromosomes are generated as a result of telomeric fusions [142] or form due to the presence of double strand breaks [143]. Repetitive DNA replications leads to telomere erosion and thus the formation of sticky ends on the chromosome. Chromosomal bridges generated by the fusions of telomeric regions of different chromosomes lead to cytokinetic failures, hence aneuploidy and cells bearing extra centrosomes [144-146]. Independent studies have discussed the correlation between centrosome amplification status and aneuploidy caused by telomere erosion. To clarify the origin of genome instability, researchers investigated centrosome amplification in different multiple myeloma stages, a cancer type well characterized by aneuploidy. They highlighted that centrosome amplification is frequently present even in early stages of myeloma. Moreover, they reported that there is no significant difference between ploidy subtypes (hyperdiploid: whole chromosome gains/losses; non-hyperdiploid: structural abnormalities and translocations) in terms of centrosome amplification [147].  This implied that telomere fusions and centrosome amplification contribute to CIN separately in the case of multiple myeloma. In another study, it was reported that telomere dysfunction and p16INK4a deficiency in breast cancer cooperatively caused centrosomal aberrations in both diploid and polyploid cells, even in the presence of functional p53 [148].  A novel molecular mechanism between telomeres and centrosomes was recently identified by another group. Telomerase transcriptional element interacting factor (TEIF), primarily known for its function in hTERT activation, was shown to localize to centrosomes in all stages of the cell cycle, and centrosomal recruitment was increased by telomere dysfunction [149]. In addition, EGF/PI3K signalling was identified as an important regulator of centrosomal recruitment of TEIF, resulting in centrosome amplification [150]. In colorectal cancers, a positive correlation between TEIF expression and centrosome amplification was reported [151], suggesting that TEIF could be playing a crucial role mediating telomere dysfunction and centrosome aberrations.”
